# *Tricholoma matsutake*-Derived Peptides Ameliorate Inflammation and Mitochondrial Dysfunction in RAW264.7 Macrophages by Modulating the NF-κB/COX-2 Pathway

**DOI:** 10.3390/foods10112680

**Published:** 2021-11-03

**Authors:** Mengqi Li, Qi Ge, Hanting Du, Songyi Lin

**Affiliations:** National Engineering Research Center of Seafood, School of Food Science and Technology, Dalian Polytechnic University, Dalian 116034, China; lmqi912@163.com (M.L.); ab1942791040@163.com (Q.G.); Duhanting960415@163.com (H.D.)

**Keywords:** *Tricholoma matsutake*, RAW264.7 macrophages, inflammation, NF-κB, mitochondrial dysfunction

## Abstract

*Tricholoma matsutake* is an edible fungus that contains various bioactive substances, some of them with immunostimulatory properties. Presently, there is limited knowledge about the functional components of *T. matsutake*. Our aim was to evaluate the protective effects and molecular mechanisms of two *T. matsutake*-derived peptides, SDLKHFPF and SDIKHFPF, on lipopolysaccharide (LPS)-induced mitochondrial dysfunction and inflammation in RAW264.7 macrophages. *Tricholoma matsutake* peptides significantly ameliorated the production of inflammatory cytokines and inhibited the expression of COX-2, iNOS, IKKβ, p-IκB-α, and p-NF-κB. Immunofluorescence assays confirmed the inhibitory effect of *T. matsutake* peptides on NF-κB/p65 nuclear translocation. Furthermore, the treatment with *T. matsutake* peptides prevented the accumulation of reactive oxygen species, increased the Bcl-2/Bax ratio, reversed the loss of mitochondrial membrane potential, and rescued abnormalities in cellular energy metabolism. These findings indicate that *T. matsutake* peptides can effectively inhibit the activation of NF-κB/COX-2 and may confer an overall protective effect against LPS-induced cell damage.

## 1. Introduction

Inflammation is a complex biological process triggered in response to various external stimuli. However, excessive and persistent inflammation can lead to diseases [1]. Macrophages play a key role in natural immunity and exhibit various functions, such as phagocytosis, antigen presentation, and cytokine secretion, in inflammatory diseases [2]. Stimulation of macrophages by endotoxin lipopolysaccharides (LPS) elicits multiple inflammatory responses, activates intracellular signaling cascades, and promotes the release of large amounts of pro-inflammatory cytokines [3]. Several key disease pathogenesis factors, including inflammation, mitochondrial dysfunction, protein folding dysfunction, oxidative stress, and programmed cell death, have been proposed as contributing to disease progression [4].

A previous study showed that an imbalance in intracellular antioxidant enzymes is conducive to the accumulation of reactive oxygen species (ROS) in LPS-induced inflammation [5]. Of note, ROS are highly reactive and can damage cellular components such as lipids, nucleic acids, proteins, and organelles. Mitochondria are the main source of intracellular ROS, making them susceptible to oxidative damage [6]. Excessive ROS can cause depolarization of the mitochondrial membrane and induce oxidative stress [7]. Mitochondria, which are unique double-membrane subcellular organelles, provide energy for the body, mediate certain metabolic functions, and encode a limited number of genes. Mitochondrial damage can lead to cell death and, in multicellular organisms, organ damage [8]. Nuclear factor kappa B (NF-κB) is a transcription factor in eukaryotic cells. It mediates inflammatory and immune responses by regulating the expression of related factors such as cyclooxygenase-2 (COX-2), as well as apoptosis through transcriptional regulation of genes, including members of the B-cell lymphoma-2 (Bcl-2) family [9]. Notably, COX-2 is also associated with apoptosis under stressful conditions [10].

*Tricholoma matsutake* (S. Ito and S. Imai) Singer is a highly appreciated edible fungus, rich in protein and amino acids. It contains important bioactive substances, such as double-chain polysaccharides, polypeptides, and matsutakeols, and has been associated with anti-tumor, antiviral, anti-diabetic, hepatoprotective, and immunostimulatory activity [11]. Food-derived bioactive peptides are attracting increasing attention due to their proven safety, limited side effects, and multiple beneficial biological functions. Peptides from edible mushrooms are gaining increasing interest due to their immunostimulatory, hormone-modulating, antibacterial, and antiviral properties, while certain plant components have been shown to have the ability to inhibit the production of inflammatory cytokines [12]. At present, however, there is limited knowledge about the functional components of *T. matsutake*. In a previous study, we identified two peptides, Ser-Asp-Leu-Lys-His-Phe-Pro-Phe (SDLKHFPF) and Ser-Asp-Ile-Lys-His-Phe-Pro-Phe (SDIKHFPF), from *T. matsutake* protein hydrolysate by simulating gastrointestinal digestion in vitro and demonstrated their efficacy in reducing intestinal inflammation in mice with DSS-induced colitis [13,14].

The objective of the present study was to assess the anti-inflammatory activity of *T. matsutake*-derived peptides SDLKHFPF and SDIKHFPF in LPS-induced RAW264.7 macrophages and particularly their protective effect against mitochondrial dysfunction. Moreover, to obtain a mechanistic insight, this study sought to analyze the intracellular signaling pathways activated by these peptides.

## 2. Material and Methods

### 2.1. Materials

LPS (*Escherichia coli* 055: B5) were provided by Sigma-Aldrich Chemicals Co. (St. Louis, MO, USA). Fetal bovine serum (FBS) was provided by PAN-Biotech GmbH (Aidenbach, Germany). Penicillin–streptomycin (P/S) and Dulbecco’s modified Eagle’s medium (DMEM) were provided by Gibco BRL Life Technology (Gibco BRL, Gaithersburg, MD, USA). Bicinchoninic acid (BCA), ROS, and Annexin V-FITC/PI double staining cell apoptosis assay kits were obtained from the Nanjing Jiancheng Bioengineering Institute (Nanjing, China). Mitochondrial membrane potential (MMP) and BeyoECL Moon assay kits were purchased from Beyotime Biotechnology Co., Ltd. (Shanghai, China). Seahorse XFp Cell Mito Stress Test kit (cat. no. 103017) and Glycolysis Stress Test kit (cat. no. 103010) were obtained from Seahorse Bioscience Company (Billerica, MA, USA). Polyvinylidene fluoride (PVDF) membrane was obtained from Millipore Corporation (Bedford, MA, USA). Primary antibodies against COX2 (ab62331,1:1000), inducible nitric oxide synthase (iNOS; ab178945, 1:1000), NF-κB p65 (ab32536, 1:100), IKKβ (ab32135, 1:2000), IκB-α (ab32518, 1:5000), phospho-IκB-α (ab133462, 1:10,000), Bax (ab32503, 1:5000), and Bcl-2 (ab182858, 1:2000) were obtained from Abcam (Cambridge, UK). Primary antibodies against phospho-NF-κB p65 (Ser536; 3033, 1:1000) and β-actin (4970, 1:1000) were provided by Cell Signaling Technology (Danvers, MA, USA). Secondary antibodies (111-035-003, 1:10,000) were purchased from Jackson ImmunoResearch Inc. (West Grove, PA, USA). Other chemicals and reagents are indicated in the specified methods. Antifading mounting medium (containing 4′,6-diamidino-2-phenylindole, DAPI) was purchased from Solarbio Science & Technology Co., Ltd. (Beijing, China).

### 2.2. Cell Cultures

RAW264.7 macrophages were obtained from the Cell Resource Center of Chinese Academy of Sciences (Shanghai, China). The cells were cultured in the DMEM complete medium containing a mixture of 10% FBS and 1% P/S at 37 °C in a 5% CO_2_ incubator.

### 2.3. Inflammatory Stimulation

RAW264.7 macrophages were seeded onto microplates and incubated for 24 h, followed by pretreatment with the respective peptide solution dissolved in the DMEM complete medium for 2 h and then co-incubation with LPS (1 μg/mL) for 24 h. The peptide replaced by the DMEM complete medium was the blank control, while LPS was selected as a positive control.

### 2.4. Griess Reagent Assay

RAW264.7 macrophages were seeded onto 96-well plates (10^5^ cells/mL) and treated as described in Section 2.3. In addition, 50 μL of cell media was collected and incubated with 50 μL of Griess reagent I and Griess reagent II for 15 min. The OD value was read at 540 nm. The NO concentration was calculated with a standard curve from sodium nitrite.

### 2.5. Cytokine Assay

RAW264.7 macrophages were seeded onto 24-well plates (10^5^ cells/mL) and treated as described in Section 2.3. The levels of interleukin (IL)-6 and tumor necrosis factor (TNF)-α in the cell culture supernatant were determined using Enzyme Linked Immunosorbent Assay (ELISA) kits (Nanjing Jiancheng Bioengineering Institute, Nanjing, China) according to the manufacturer’s protocols.

### 2.6. Intracellular ROS Assay

RAW264.7 macrophages were seeded onto 6-well plates (10^5^ cells/mL) and treated as described in Section 2.3. The 2′,7′-dichlorodihydrofluorescein diacetate (DCFH-DA) probe at a final concentration of 10 μM was added to the cells and incubated at 37 °C for 40 min. Then, the cells were washed three times with phosphate buffered saline (PBS) and visualized using an inverted fluorescence microscope (Carl Zeiss Jena GmbH, Jena, Germany).

### 2.7. Flow Cytometric Analysis of Cell Apoptosis

RAW264.7 macrophages were seeded onto 6-well plates (10^5^ cells/mL) and treated as described in Section 2.3. Apoptosis was evaluated by double staining with Annexin V-FITC/PI following the manufacturer’s protocol. After washing with PBS, the supernatant was discarded, and the cells were collected and resuspended in a 500 μL binding solution. Then, the cells were gently mixed with 5 μL Annexin V-FITC and 5 μL PI. The mixture was incubated in the dark at room temperature for 10 min and then analyzed using flow cytometry (BD Biosciences, San Jose, CA, USA).

### 2.8. Imaging of MMP

RAW264.7 macrophages were seeded onto 12-well plates (10^5^ cells/mL) and treated as described in Section 2.3. The cells were washed three times with PBS, three times with PBS, and stained with the JC-1 fluorescent probe as indicated by the MMP analysis kit. Briefly, 1 mL of JC-1 working solution was added to 1 mL of cell culture medium and incubated at 37 °C for 20 min. Fluorescence images were captured by an inverted microscope.

### 2.9. Oxygen Consumption Rate (OCR) and Extracellular Acidification Rate (ECAR) Assay

RAW264.7 macrophages were seeded onto XF 8-well plates (10^4^ cells/well) and treated as described in Section 2.3. OCR and ECAR were measured in real-time using the Seahorse XFp Cell Mito Stress Test kit and Glycolysis Stress Test kit with an XFp Extracellular Flux Analyzer (Agilent Technologies, Santa Clara, CA, USA). Following the manufacturer’s protocols, OCR was assayed by sequentially adding 20 µL oligomycin (oligo; 15 μM), 22 µL carbonyl cyanide-4 (trifluoromethoxy) phenylhydrazone (FCCP; 10 μM), and 25 µL rotenone/antimycin A (Rote/AA; 5 µM) to the plates at the indicated times. Likewise, ECAR was measured by sequentially adding 22 µL glucose (Glu; 100 mM), 20 µL oligo (10 mM), and 25 µL 2-deoxyglucose (2-DG; 500 mM) at the indicated times.

### 2.10. Western Blotting

RAW264.7 macrophages were seeded onto 6-well plates (10^5^ cells/mL) and treated as described in Section 2.3. Total proteins were extracted in RIPA lysis buffer containing 1% phenylmethylsulfonyl fluoride protease inhibitor, and protein concentrations were determined using a BCA assay kit. Denatured proteins were separated using 12% sodium dodecyl sulphate–polyacrylamide gel electrophoresis gels and transferred to PVDF membranes. Membranes were blocked with 5% non-fat milk for 1 h and then incubated with primary antibodies overnight at 4 °C, followed by incubation with horseradish peroxidase-conjugated secondary antibody at room temperature for 1 h. Finally, the BeyoECL Moon detection reagent was used to visualize the bands and Image J software was used for their quantification.

### 2.11. Immunofluorescence Assay

RAW264.7 macrophages were seeded (10^4^ cells/mL) onto 12-well plates and treated as described in Section 2.3. The supernatant was discarded, and cells were fixed with 4% paraformaldehyde solution. Then, 0.1% Triton-X 100 was added for cell permeabilization. Subsequently, cells were blocked by the addition of goat serum and incubated with rabbit anti-mouse NF-κB/p65 antibody (1:200) overnight at 4 °C. The following day, the cells were incubated with fluorescein isothiocyanate (FITC)-labeled fluorescent secondary antibodies for 1 h at room temperature. Then, 1–2 drops of a mounting medium containing DAPI were added, and images were captured on an inverted microscope. The quantitative analysis was performed using ImageJ software.

### 2.12. Statistical Analysis

The variance of data was carried out employing the SPSS 22 software (SPSS Inc., Chicago, IL, USA) and Origin 2018 software (OriginLab Corporation, Northampton, MA, USA). All of the experiments were performed in triplicate, and data were expressed as means ± standard deviations. *p* < 0.05 was represented for the significant differences and the results were tested by one-way ANOVA and Duncan’s multiple tests.

## 3. Results and Discussion

### 3.1. Tricholoma matsutake-Derived Peptides Suppress the LPS-Induced Inflammatory Response in RAW264.7 Macrophages

Inflammation is a complex biological process regulated by a range of inflammatory mediators and cytokines [2]. Nitric oxide (NO) is an important inflammatory mediator, and excess NO has been reported to react with superoxide anions to generate peroxide nitrite, thereby promoting the occurrence of inflammatory diseases [15]. As demonstrated by the Griess reagent assay results, NO production increased dramatically in LPS-stimulated RAW264.7 macrophages, but the trend was reversed by the *T. matsutake* peptide pretreatment (Figure 1A). All of the concentrations of SDIKHFPF and SDLKHFPF (50, 100, and 200 μg/mL) significantly inhibited the LPS-induced NO overproduction. In addition, we observed that reduction or inhibition of the activation of COX-2 regulated the synthesis and release of the iNOS and NO inflammatory mediators, thus effectively controlling the development of inflammation [16]. Interestingly, iNOS is the main enzyme that catalyzes the production of NO in LPS-treated cells. Transcription of the iNOS gene is known to be mainly regulated by the NF-κB transcription factor [17]. Furthermore, we determined the expression of iNOS and COX-2 proteins in LPS-induced RAW264.7 macrophages. Accordingly, we observed the LPS-induced overexpression of iNOS and COX-2 proteins, consistent with the production of NO, whereas this effect was significantly attenuated in cells pretreated with *T. matsutake* peptides (Figure 1B). These results suggest that SDIKHFPF and SDLKHFPF might reduce the levels of NO by inhibiting the NF-κB pathway.

### 3.2. Tricholoma matsutake-Derived Peptides Reverse LPS-Induced Intracellular Production of ROS in RAW264.7 Macrophages

Various studies have shown that pro-inflammatory cytokines can induce inflammatory responses and accelerate disease progression [18]. To investigate the effects of SDIKHFPF and SDLKHFPF on the release of inflammatory cytokines in LPS-stimulated RAW264.7 macrophages, we measured the levels of IL-6 and TNF-α using ELISA (Figure 1C,D). We found that these levels were significantly higher in LPS-induced cells than in controls, but were restored in a dose-dependent manner upon pretreatment with SDIKHFPF and SDLKHFPF. In particular, at the highest concentration of 200 µg/mL, both SDIKHFPF and SDLKHFPF were shown to significantly reduce the overproduction of IL-6 (303.03 ± 4.95 pg/mL vs. 298.99 ± 7.56 pg/mL) and TNF-α (200.57 ± 5.69 pg/mL vs. 180.46 ± 5.33 pg/mL). A previous study demonstrated that elevated levels of TNF-α were closely associated with inflammatory diseases, while overlapping and synergistic activities with the NF-κB pathway members were responsible for regulating IL-6, COX-2, and iNOS [19]. Taken together, our results demonstrate the anti-inflammatory activity of SDIKHFPF and particularly SDLKHFPF in LPS-induced RAW264.7 macrophages.

Peptides and other biological substances can alleviate the immune dysfunction caused by oxidative stress by upregulating the intracellular antioxidant enzyme system [20]. A previous study highlighted the pro-inflammatory effects of ROS [5]. Here, LPS-stimulated intracellular ROS were quantified using the DCFH-DA green fluorescent probe. As shown in Figure 2A, the fluorescence intensity of RAW264.7 macrophages increased significantly (*p* < 0.05) after exposure to LPS compared with the control group, indicating that LPS induced the accumulation of ROS. However, pretreatment with the *T. matsutake* peptides SDLKHFPF and SDIKHFPF effectively attenuated fluorescence intensity compared with the cells stimulated with LPS only. Analysis of the mean fluorescence intensity of RAW264.7 macrophages (Figure 2B) revealed that both peptides significantly inhibited intracellular ROS accumulation, leading to 30.11% and 61.13% fewer ROS with SDIKHFPF and SDLKHFPF, respectively, compared with the cells exposed to LPS alone. The critical factors that affect the physiological activity of different peptides are amino acid composition, amino acid sequence, and chemical properties such as hydrophobicity [21]. Similar results were reported by Qian et al. [22] concluding that the antioxidative activity of a peptide depends on its molecular size and chemical properties, such as the hydrophobicity and electron transfer capacity of the amino acid residues in the sequence. Numerous studies have shown that ROS increase dramatically following the stimulation of the cell membrane of phagocytes, which in turn triggers inflammation through a series of oxidative stress-affected signaling transduction pathways [16,23]. Therefore, the synthetic peptides SDIKHFPF and SDLKHFPF protect cells from highly reactive oxidation products, which can cause damage to biomolecules.

### 3.3. Tricholoma matsutake-Derived Peptides Rescue LPS-Induced MMP Loss in RAW264.7 Macrophages

The excessive accumulation of ROS impairs the mitochondrial function. MMP originates from the asymmetric distribution of protons and other ions on both sides of the inner membrane of mitochondria during respiratory oxidation. In addition, its stability facilitates the maintenance of the normal physiological functions of cells [24]. Here, we examined the changes in MMP in LPS-induced RAW264.7 macrophages using the JC-1 fluorescent probe. JC-1 aggregated in the mitochondrial matrix, resulting in a strong red fluorescent signal and weak green fluorescence in control RAW264.7 macrophages (Figure 3A). This result indicated a high MMP. In contrast, when RAW264.7 macrophages were stimulated with LPS, JC-1 was converted to the monomer form, producing a strong green fluorescence and weak red fluorescence. This behavior suggested that LPS depolarized the mitochondrial membrane. SDIKHFPF and SDLKHFPF promoted JC-1 aggregation in the mitochondrial matrix, reversing the loss of MMP in LPS-stimulated RAW264.7 macrophages. Furthermore, the fluorescence intensity ratio of JC-1 aggregates to JC-1 monomers confirmed that both SDIKHFPF and SDLKHFPF exerted a significant and similar effect (Figure 3B). Zhang et al. [25] reported that excessive ROS could cause mitochondrial damage by decreasing MMP. Moreover, the latter is a landmark event in the early stages of apoptosis [26]. Our results indicated that pretreatment with *T. matsutake* peptides could effectively protect the mitochondrial membrane from depolarization by preventing intracellular ROS accumulation, revealing a beneficial effect on LPS-induced damage.

### 3.4. Tricholoma matsutake-Derived Peptides Attenuate LPS-Induced Apoptosis in RAW264.7 Macrophages

We evaluated the effect of *T. matsutake* peptides on the apoptosis rate in LPS-stimulated RAW264.7 macrophages by flow cytometry using Annexin V in combination with propidium iodide to distinguish cells at different stages of apoptosis (Figure 4A). Consistent with the results of Yuan et al., [27] the proportion of apoptotic cells increased from 2.73 ± 1.14% to 19.77 ± 1.35% after LPS stimulation (Figure 4B), but was maintained at 13.43 ± 1.07% and 9.27 ± 1.11%, upon SDIKHFPF and SDLKHFPF pretreatment, respectively. Shan et al. [28] suggested that ROS regulated intracellular signaling cascades, thereby activating a series of programmed cell death pathways. Afrin et al. [29] reported that the LPS treatment induced apoptosis of RAW264.7 macrophages through secretion of NO and TNF-α and activation of different molecular pathways. COX-2 and its products are implicated in apoptosis during the pathogenesis of many diseases [30]. In this study, the elevated rate of apoptosis in the LPS-treated group may be attributed to high concentrations of pro-inflammatory mediators and oxidative damage. The *T. matsutake* peptide treatment had the opposite effect, lowering the level of pro-inflammatory mediators and, consequently, apoptosis. Similar results reported by Yi et al. [31] showed that soybean-derived peptides inhibited MG132-induced apoptosis of RAW264.7 cells in a dose-dependent manner by flow cytometry. In summary, our results indicate that the *T. matsutake* peptide pretreatment significantly attenuates the LPS-induced apoptotic cascade in RAW264.7 macrophages.

### 3.5. Tricholoma matsutake-Derived Peptides Improve LPS-Induced Mitochondrial Respiration Dysfunction in RAW264.7 Macrophages

Cellular energy metabolism is an essential process in all organisms. Under normal physiological conditions, macrophages rely on glucose and oxidative phosphorylation (OXPHOS) to meet their energy requirements. OXPHOS is carried out in mitochondria through activation of the electron transport chain (ETC). However, when macrophages are stimulated by pathogens such as LPS or by potential inflammasomes, metabolic reprogramming may occur [32,33]. Here, we used the XFp Extracellular Flux Analyzer to determine several parameters of mitochondrial respiration and glycolysis and thus assessed the effect of *T. matsutake* peptides on mitochondrial functionality. Injection of oligomycin inhibited ATP synthase and reduced ATP production (Figure 5A), leading to a decrease in mitochondrial respiration. Proton leakage can be used as a marker of mitochondrial damage. We injected the FCCP uncoupling agent to reveal the maximal respiration capacity (MRC) achievable by the cells. The portion of MRC that exceeds basal respiration is defined as the spare respiration capacity (SRC). It represents the ability to address energy demands and can be used as an indicator of cell health or flexibility. An injection of rotenone/antimycin A was used to turn off mitochondrial respiration. As depicted in Figure 5B, the LPS treatment significantly reduced basal respiration, MRC, proton leakage, and SRC compared with the control group. This response was mainly due to the loss of ATP-linked respiration in mitochondria. Importantly, SDIKHFPF and SDLKHFPF treatments increased ATP-linked respiration to 78.38 ± 0.20 pmol/min and 91.33 ± 1.59 pmol/min (*p* < 0.05), respectively, compared with the model cells. Furthermore, the SDIKHFPF treatment increased basal respiration, MRC, proton leakage, and SRC to 101.41 ± 0.34, 139.01 ± 0.83, 23.02 ± 0.14, and 37.61 ± 0.49 pmol/min, respectively, whereas the SDLKHFPF addition achieved 112.94 ± 3.74, 158.49 ± 3.58, 21.60 ± 2.15, and 45.55 ± 0.16 pmol/min, respectively.

To evaluate ECAR (Figure 5B), glucose was added after the energy generated through glycolysis was depleted. Glucose-induced responses were reported as the rate of glycolysis under basal conditions. We injected oligo to shift the ATP source to glycolysis, with the subsequent increase in ECAR indicating the maximum cellular glycolytic capacity. The difference between glycolytic capacity and glycolysis rate defined the glycolytic reserve, and glycolysis was inhibited using 2-DG. Compared with controls, the cells exposed to LPS exhibited significantly increased glycolysis and glycolytic capacity. However, this effect was offset to a certain extent by the pretreatment with *T. matsutake* peptides. We observed that the treatment with both SDIKHFPF and SDLKHFPF significantly reduced glycolysis (86.05 ± 1.76 vs. 81.37 ± 7.76 mpH/min) and glycolytic capacity (126.24 ± 0.64 vs. 120.15 ± 2.39 mpH/min) compared with the model cells. Moreover, we found that the glycolytic reserve was decreased (40.19 ± 1.12 and 38.78 ± 10.15 mpH/min), although no significant differences were observed in the values relative to those observed upon the treatment with LPS alone. The extracellular flux analysis showed that LPS significantly interfered with the energy metabolism in RAW264.7 macrophages, which was further supported by the increase in ECAR and the decrease in OCR. However, the treatment with *T. matsutake* peptides tended to switch energy production from glycolysis to aerobic metabolism, suggesting a beneficial role of the peptides in attenuating the effect of LPS on the metabolic disorders of RAW264.7 macrophages. Mills et al. [34] suggested that LPS stimulated macrophages to activate and release NO, which might impair the ETC and thereby reduce the efficiency of OXPHOS. Usually, a drop in ATP indicates that the mitochondrial function is compromised or decreased, which may lead to apoptosis [35]. Since MMP drives the ATP synthesis via OXPHOS, any changes in the former can block ATP production. When the mitochondrial function is impaired, the OXPHOS capacity is reduced, disrupting antioxidant defenses [36]. Liu and Ho [37] found that LPS stimulated macrophage activation, leading to increased glucose uptake and accelerated anaerobic glycolytic processes. Similar results reported by Zhao et al. [38] indicated that sea cucumber ovum peptide NDEELNK had a protective effect on the mitochondrial energy metabolism disorder caused by scopolamine damage to PC12 cells. Our current results are consistent with these reports. *T. matsutake* peptides can reduce LPS-induced intracellular ROS levels and apoptosis in RAW264.7 macrophages, reverse MMP loss and abnormal cellular metabolism, and have a protective effect on LPS-induced cell damage.

### 3.6. Effect of Tricholoma matsutake-Derived Peptides on the NF-κB/COX-2 Signaling Pathway in LPS-Induced RAW264.7 Macrophages

The NF-κB/p65 protein, as the core regulator of inflammatory genes, is known to promote the expression of inflammatory cytokines. Both the activation of the NF-κB signaling pathway and nuclear translocation of NF-κB/p65 are important events in the inflammatory response [13]. We examined the protein expression levels of IKKβ, IκB-α, p-IκB-α, and p-NF-κB/p65 in RAW264.7 macrophages using the Western blot analysis. A previous study showed that the activation of the NF-κB signaling pathway depended mainly on phosphorylation of the IκB kinase (IKK), and particularly its IKKβ subunit [4]. Western blotting revealed that, compared with the control group, administration of LPS alone resulted in significant upregulation of IKKβ and p-IκB-α in RAW264.7 macrophages (Figure 6A). This pattern is in line with the findings of Hoesel and Schmid [39] that IκB-induced phosphorylation of IκB-α led to degradation of IκB-α and activation of NF-κB, thereby promoting the expression of COX-2, iNOS, and TNF-α. Consistent with our results regarding the levels of pro-inflammatory cytokines, we noticed that the suppression of NF-κB was accompanied by a reduction in the production of cytokines. Furthermore, we found that the expression of the phosphorylated-NF-κB/p65 protein was upregulated in LPS-stimulated RAW264.7 macrophages compared with the untreated controls, confirming the activation of the NF-κB pathway. The pretreatment with *T. matsutake* peptides significantly improved the expression of these proteins. The above results indicated that the pretreatment with *T. matsutake* peptides could significantly suppress the activation of the NF-κB pathway. As revealed by immunofluorescence staining, LPS induced the nuclear distribution of NF-κB/p65, whereas the SDIKHFPF and SDLKHFPF pretreatment significantly decreased it (Figure 6B). Therefore, *T. matsutake* peptides effectively reversed the translocation of NF-κB/p65 from the cytoplasm to the nucleus. Taken together, these results indicate that SDIKHFPF and SDLKHFPF peptides can partially inhibit the activation of the NF-κB signaling pathway by suppressing the degradation of IκBα and blocking the nuclear translocation of NF-κB/p65 in LPS-induced RAW264.7 macrophages. Similarly, Ren et al. [18] proved that the hazelnut protein-derived peptide LDAPGHR exerts anti-inflammatory effects by inhibiting the LPS-induced activation of NF-κB and MAPK pathways in RAW264.7 macrophages. *T. matsutake*-derived peptides might exert anti-inflammatory effects by directly scavenging or neutralizing ROS or by inhibiting their downstream adverse effects. Mitochondria participate in the intrinsic apoptotic pathway, causing the activation of the caspase signaling pathway upon attack by ROS [40]. The Bcl-2 family plays a crucial role in apoptosis, wherein Bcl-2 acts as an anti-apoptotic regulator, whereas Bax acts as a pro-apoptotic regulator [41,42]. Furthermore, we investigated the effect of *T. matsutake* peptides on the expression of these two important apoptosis-related proteins, Bcl-2 and Bax. We found that LPS reduced the expression of Bcl-2, whereas the pretreatment with SDIKHFPF and SDLKHFPF significantly attenuated this effect. Conversely, we observed that LPS promoted the expression of Bax, whereas the pretreatment with SDIKHFPF and SDLKHFPF inhibited its expression. The Bcl-2/Bax ratio was shown to be negatively correlated with the apoptosis rate. In particular, the quantitative analysis showed that LPS reduced the Bcl-2/Bax ratio, while SDIKHFPF and SDLKHFPF partially reversed the effect of LPS (Figure 6A). Song et al. [10] found that the activation of NF-κB directly decreases the Bcl-2/Bax ratio, which is known to be related to an increase in apoptosis. Cai and Harrison [43] reported that excessive pathological levels of NO could also trigger cell apoptosis. In addition, NF-κB can regulate the Bcl-2/Bax ratio in an indirect manner via COX-2 [10], which is consistent with our findings on COX-2 protein expression. Therefore, we hypothesize that *T. matsutake* peptides can effectively improve LPS-induced inflammation and mitochondrial dysfunction in RAW264.7 macrophages by inhibiting the activation of NF-κB and expression of COX-2.

## 4. Conclusions

In conclusion, we demonstrated that *T. matsutake*-derived peptides SDLKHFPF and SDIKHFPF attenuate the inflammatory response of LPS-induced RAW264.7 macrophages by blocking the NF-κB/COX-2 signaling pathway, exhibiting a protective effect against cell mitochondrial dysfunction. Therefore, the *T. matsutake* peptide can be used as a potential natural food source for the reduction of the severity of inflammatory diseases.

## Figures and Tables

**Figure 1 foods-10-02680-f001:**
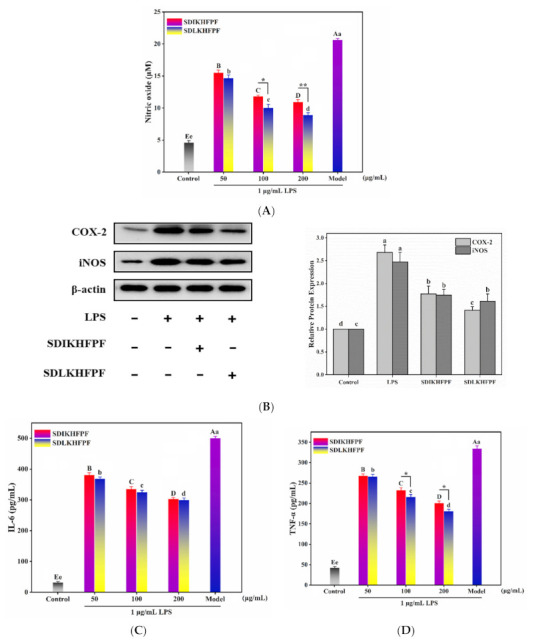
(**A**) Determination of NO production. (**B**) Protein expression of COX-2 and iNOS determined by Western blotting. β-actin was used as a loading control. (**C**,**D**). Determination of IL-6 and TNF-α contents in the medium supernatants. All of the data shown were the mean ± SD (*n* = 3), *: *p* < 0.05, **: *p* < 0.01. The different capital letters represent the significant difference for SDIKHFPF (*p* < 0.05). The different lowercase letters represent the significant difference for SDLKHFPF (*p* < 0.05).

**Figure 2 foods-10-02680-f002:**
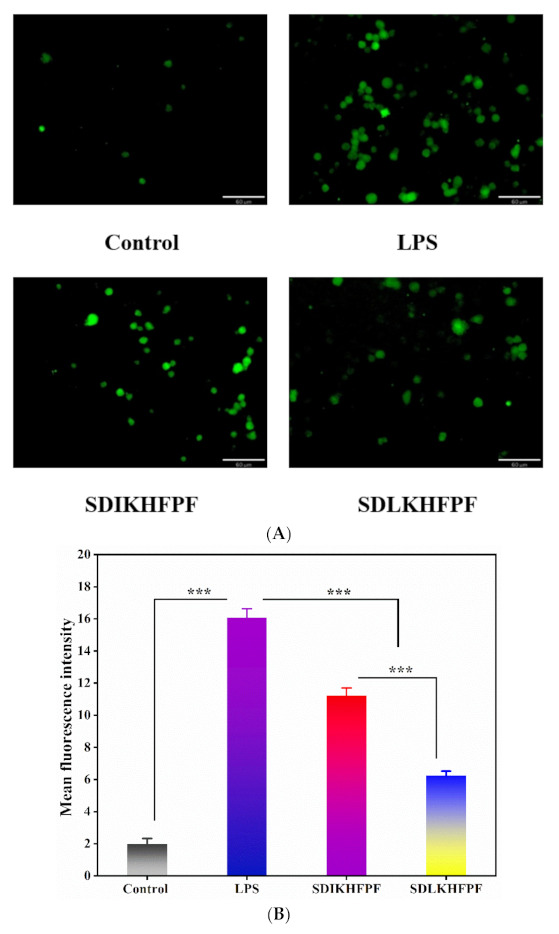
(**A**) Fluorescence inversion microscopy imaging of ROS in RAW264.7 macrophages. ROS labeled with DCFH-DA (green: DCFH-DA). (**B**) The mean fluorescence density of RAW264.7 macrophages. Data were represented as mean ± SD (*n* = 3), ***: *p* < 0.001.

**Figure 3 foods-10-02680-f003:**
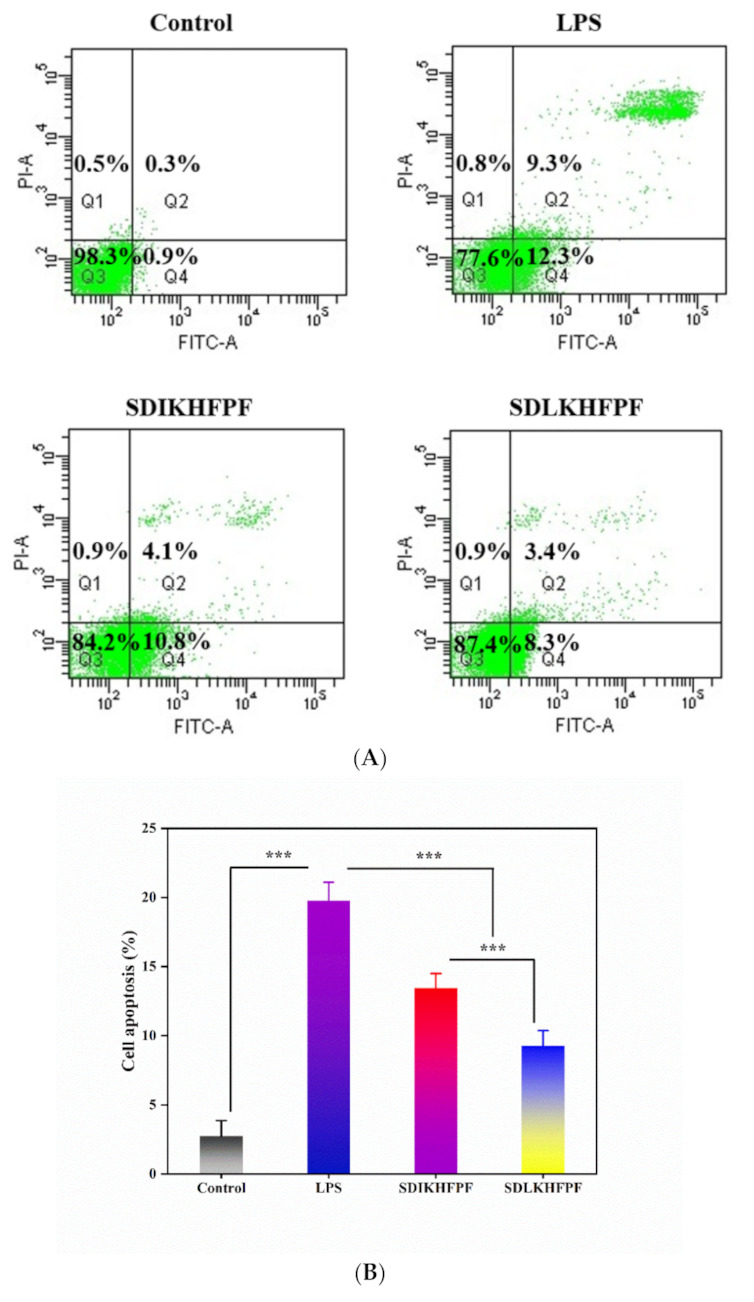
(**A**) Apoptosis was measured with annexin V/PI staining combined with flow cytometry. (**B**) Quantitative analysis of apoptosis was performed by combining the percentages of early (Q2) and late (Q4) apoptosis. Data were represented as mean ± SD (*n* = 3), ***: *p* < 0.001.

**Figure 4 foods-10-02680-f004:**
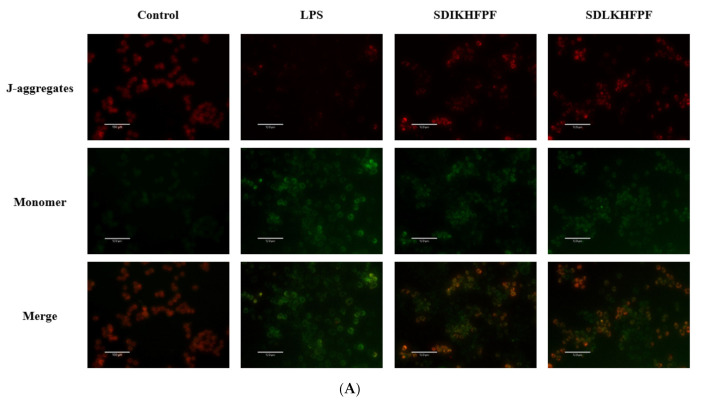
(**A**) Fluorescence microscopy imaging of the mitochondrial membrane potential for RAW264.7 macrophages stained by JC-1. (**B**) The mean fluorescence density of red/green fluorescence intensity ratio. Data were represented as mean ± SD (*n* = 3), ***: *p* < 0.001.

**Figure 5 foods-10-02680-f005:**
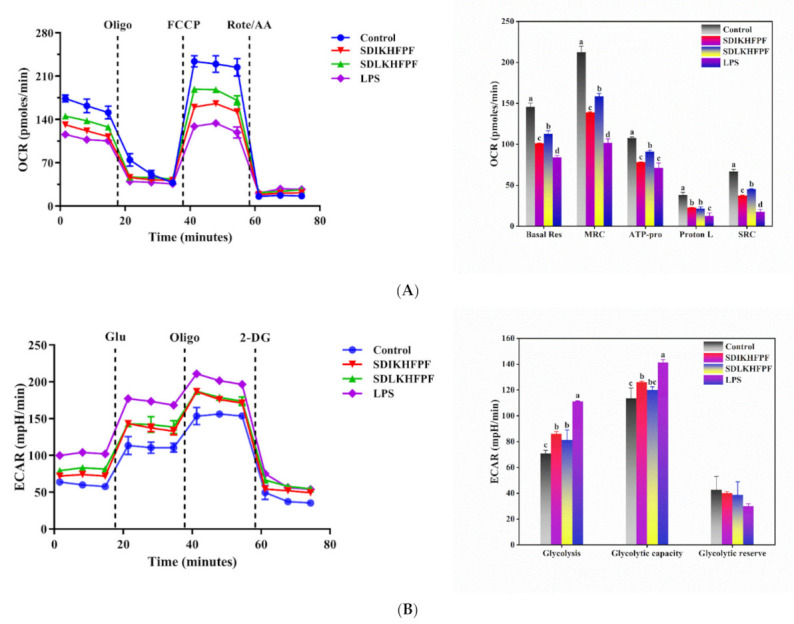
(**A**) Regulation of mitochondrial respiration by T. matsutake-derived peptides in LPS-induced RAW264.7 macrophages. The basal respiration (Basal Res), ATP production (ATP-pro), proton leak (Proton L), maximal respiration capacity (MRC), and spare respiration capacity (SRC) were calculated from the XF cell Mito stress test profile. (**B**) Regulation of glycolysis by T. matsutake-derived peptides in LPS-induced RAW264.7 macrophages. The glycolysis, glycolytic capacity, and glycolytic reserves were calculated from the XF glycolysis stress test profile. All of the data are indicated as the mean ± SD (*n* = 3). Different letters mean statistically significant differences at the level of *p* < 0.05.

**Figure 6 foods-10-02680-f006:**
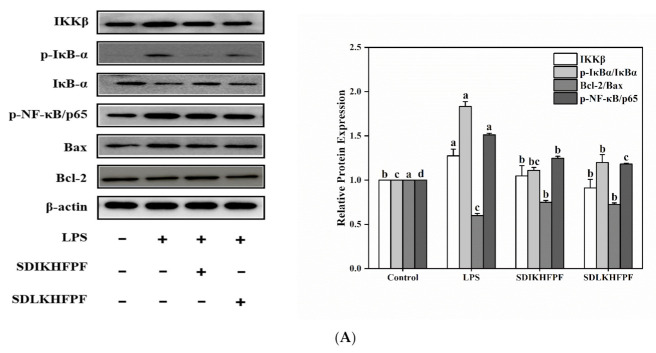
(**A**) The Western blot analysis of IKKβ, p-IκBα, Bcl-2/Bax, and p-NF-κB/p65 in RAW264.7 macrophages. β-actin was used as a loading control. (**B**) The localization of p65 in the cytoplasm and nucleus was measured by immunofluorescence staining. Blue fluorescence represents cell nucleus and green fluorescence represents p65. All of the data shown were the mean ± SD (*n* = 3). Different letters mean statistically significant differences at the level of *p* < 0.05.

## Data Availability

The data showed in this study are contained within the article.

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
