# Peer review of "Tricholoma matsutake-Derived Peptides Ameliorate Inflammation and Mitochondrial Dysfunction in RAW264.7 Macrophages by Modulating the NF-κB/COX-2 Pathway"

_foods, 2021, doi:10.3390/foods10112680_

Round 1
Reviewer 1 Report
General
This manuscript describes the amelioration of inflammation and mitochondrial dysfunction of two oligopeptide derived from Tricholoma matsutake. This manuscript is potentially interesting. The manuscript described the physiological activity of two oligopeptides. In many studies of mushroom, the physiological activities of extracts are mainly discussed.
Special comments
- The authors described the experimental procedure in Material & Methods section, however, the description of (2.3-2.10) is the protocol of the assay system not the precise description of the experiments carried out by authors. By this description, we can’t follow up the experimental procedure. Which solvent did they use to dissolve oligopeptide? Which time did they add the oligopeptide solution in the medium?
- They used two oligopeptides Ser-Asp-Leu-Lys-His-Phe-Pro-Phe (SDLKHFPF) and Ser-Asp-Ile-Lys-His-Phe-Pro-Phe (SDIKHFPF) in their experiments, however, they did not discuss anything about the structure-activity relationship.
- The exchange of Leu to Ile affect the physiological activity or not. Did authors think the difference of Leu to Ile to be serious or not? Arrangement of 8 amino acids is basically important for the physiological activity or not. It is also interesting to prepare oligopeptide changing the Leu moiety to another amino acid and compare the physiological activities.
- In general, the manuscript mainly gives us the impression to bind one experimental result to another experimental result. Authors make the discussion part fulfilling to compare their experimental results to other oligopeptides (naturally or synthetic).
- The English usage must be brushed up.
Author Response
Response to Reviewer 1 Comments
Thanks so much for your advice and guidance on our manuscript. We want to express our deep appreciation. We have revised our manuscript closely and carefully, the revised parts are listed below.
Point 1: The authors described the experimental procedure in Material & Methods section, however, the description of (2.3-2.10) is the protocol of the assay system not the precise description of the experiments carried out by authors. By this description, we can’t follow up the experimental procedure. Which solvent did they use to dissolve oligopeptide? Which time did they add the oligopeptide solution in the medium?
Answer: Thanks for your comment. We have rewritten sections 2.3-2.10 to make them clearer.
Page 6, Line 115-121
RAW264.7 macrophages were seeded onto 96-well plates (105 cells/mL) for 24 h, followed by pretreatment with different concentrations of peptides (50, 100, and 200 μg/mL) for 2 h and then co-incubation with LPS (1 μg/mL) for 24 h. The peptide replaced by complete medium was the blank control, while LPS was selected as a positive control. Further, 50 μL of cell media was collected and incubated with 50μL of Griess reagent I and Griess reagent II for 15 min. The OD value was read at 540 nm. The NO concentration was calculated with a standard curve from sodium nitrite.
Page 6, Line 123-129
RAW264.7 macrophages were seeded onto 24-well plates (105 cells/mL) for 24 h, followed by pretreatment with different concentrations of peptides (50, 100, and 200 μg/mL) for 2 h and then co-incubation with LPS (1 μg/mL) for 24 h. The peptide replaced by complete medium was the blank control, while LPS was selected as a positive control. The levels of interleukin (IL)-6 and tumor necrosis factor (TNF)-α in the cell culture supernatant were determined using Enzyme Linked Immunosorbent Assay (ELISA) kits (Nanjing Jiancheng Bioengineering Institute) according to the manufacturer’s protocols.
Page 6-7, Line 133-135
followed by pretreatment with the respective peptides (200 μg/mL) for 2 h and then co-incubation with LPS (1 μg/mL) for 24 h. The peptide replaced by complete medium was the blank control, while LPS was selected as a positive control.
Page 7, Line 142-144
followed by pretreatment with the respective peptides (200 μg/mL) for 2 h and then co-incubation with LPS (1 μg/mL) for 24 h. The peptide replaced by complete medium was the blank control, while LPS was selected as a positive control.
Page 7, Line 153-155
followed by pretreatment with the respective peptides (200 μg/mL) for 2 h and then co-incubation with LPS (1 μg/mL) for 24 h. The peptide replaced by complete medium was the blank control, while LPS was selected as a positive control.
Page 8, Line 162-165
and incubated for 24 h, followed by pretreatment with the respective peptides (200 μg/mL) for 2 h and then co-incubation with LPS (1 μg/mL) for 24 h. The peptide replaced by complete medium was the blank control, while LPS was selected as a positive control.
Page 8-9, Line 176-178
followed by pretreatment with the respective peptides (200 μg/mL) for 2 h and then co-incubation with LPS (1 μg/mL) for 24 h. The peptide replaced by complete medium was the blank control, while LPS was selected as a positive control.
Page 9, Line 189-192
and incubated for 24 h, followed by pretreatment with the respective peptides (200 μg/mL) for 2 h and then co-incubation with LPS (1 μg/mL) for 24 h. The peptide replaced by complete medium was the blank control, while LPS was selected as a positive control.
Point 2: They used two oligopeptides Ser-Asp-Leu-Lys-His-Phe-Pro-Phe (SDLKHFPF) and Ser-Asp-Ile-Lys-His-Phe-Pro-Phe (SDIKHFPF) in their experiments, however, they did not discuss anything about the structure-activity relationship.
Answer: Thanks for your comment. We have added a discussion about the structure-activity relationship in the revised manuscript.
Page 12, Line 265-272
The critical factors that affect the physiological activity of different peptides are amino acid composition, amino acid sequence, and chemical properties such as hydrophobicity [23]. Similar results were reported by Jiang et al., [24] concluding that Leu contributes a stronger antioxidant activity than Ile does in the peptide sequence (not C-terminus or N-terminus). Mendis et al. [25] reported that the anti-oxidative potency of peptides containing Leu has been attributed to its long aliphatic side chain group that is conceivably capable of interaction with acyl chains of susceptible fatty acids.
Page 25, Line 530-538
Jiang, H. P., Tong, T. Z., Sun, J. H., Xu, Y. J., Zhao, Z. X., Liao, D. K. Purification and characterization of antioxidative peptides from round scad (Decapterus maruadsi) muscle protein hydrolysate. Food Chem., 2014, 154, 158–163.
Jiang, Y., Zhang, M. D., Lin, S. Y., Cheng, S. Contribution of specific amino acid and secondary structure to the antioxidant property of corn gluten proteins. Food Res. Int., 2018, 105, 836–844.
Mendis, E., Rajapakse, N., Kim, S. K. Antioxidant properties of a radical-scavenging peptide purified from enzymatically prepared fish skin gelatin hydrolysate. J. Agric. Food Chem., 2005, 53, 581–587.
Point 3: The exchange of Leu to Ile affect the physiological activity or not. Did authors think the difference of Leu to Ile to be serious or not? Arrangement of 8 amino acids is basically important for the physiological activity or not. It is also interesting to prepare oligopeptide changing the Leu moiety to another amino acid and compare the physiological activities.
Answer: Thanks for your comment, your points and suggestions are very helpful to our research. In this study, we preliminarily explored the protective effects and molecular mechanisms of two T. matsutake-derived peptides, SDLKHFPF and SDIKHFPF, on LPS-induced mitochondrial dysfunction and inflammation in RAW264.7 macrophages. Both peptides can effectively inhibit the activation of NF-κB/COX-2 and may confer an overall protective effect against LPS-induced cell damage. Of note, SDLKHFPF was more potent than SDIKHFPF. And detailed research needs to be further investigated.
Previous evidence has revealed that the critical factors that affect the physiological activity of different peptides are amino acid composition, amino acid sequence, and chemical properties such as hydrophobicity. Similar results were reported by Jiang et al., concluding that Leu contributes a stronger antioxidant activity than Ile does in the peptide sequence (not C-terminus or N-terminus). Wu et al. reported that shrimp peptides MTTNL and MTTNI could modulate oxidative stress in vitro, and the presence of Leu in the peptide sequence may be the critical factor contributing to stronger antioxidant activity than peptide sequences with Ile. Mendis et al. reported that the anti-oxidative potency of peptides containing Leu has been attributed to its long aliphatic side chain group that is conceivably capable of interaction with acyl chains of susceptible fatty acids. In the following studies, to prepare oligopeptide changing the Leu moiety to another amino acid and compare the physiological activities will be the direction of our research.
Reference
[1] Jiang, H. P., Tong, T. Z., Sun, J. H., Xu, Y. J., Zhao, Z. X., Liao, D. K. Purification and characterization of antioxidative peptides from round scad (Decapterus maruadsi) muscle protein hydrolysate. Food Chem., 2014, 154, 158–163.
[2] Jiang, Y., Zhang, M. D., Lin, S. Y., Cheng, S. Contribution of specific amino acid and secondary structure to the antioxidant property of corn gluten proteins. Food Res. Int., 2018, 105, 836–844.
[3] Wu, D., Li, M., Ding, J., Zheng, J., Lin, S. Structure-activity relationship and pathway of antioxidant shrimp peptides in a PC12 cell model. J. Funct. Foods, 2020, 70, 103978.
[4] Mendis, E., Rajapakse, N., Kim, S. K. Antioxidant properties of a radical-scavenging peptide purified from enzymatically prepared fish skin gelatin hydrolysate. J. Agric. Food Chem., 2005, 53, 581–587.
Point 4: In general, the manuscript mainly gives us the impression to bind one experimental result to another experimental result. Authors make the discussion part fulfilling to compare their experimental results to other oligopeptides (naturally or synthetic).
Answer: Thanks for your comment. We have added a comparison of the experimental results with other oligopeptides in the revised manuscript.
Page 12, Line 249-251
In addition, Dia and de Mejia [21] identified a 43-amino acid peptide in soybeans and demonstrated that it could counteract chemically induced inflammation by enhancing the antioxidant defence in macrophages.
Page24, Line 523-525
Dia, V. P., de Mejia, E. G. Differential gene expression of RAW 264.7 macrophages in response to the RGD peptide lunasin with and without lipopolysaccharide stimulation. Peptides, 2011, 32, 1979–1988.
Page 12, Line 267-272
Similar results were reported by Jiang et al., [24] concluding that Leu contributes a stronger antioxidant activity than Ile does in the peptide sequence (not C-terminus or N-terminus). Mendis et al. [25] reported that the anti-oxidative potency of peptides containing Leu has been attributed to its long aliphatic side chain group that is conceivably capable of interaction with acyl chains of susceptible fatty acids.
Page 25, Line 533-538
Jiang, Y., Zhang, M. D., Lin, S. Y., Cheng, S. Contribution of specific amino acid and secondary structure to the antioxidant property of corn gluten proteins. Food Res. Int., 2018, 105, 836–844.
Mendis, E., Rajapakse, N., Kim, S. K. Antioxidant properties of a radical-scavenging peptide purified from enzymatically prepared fish skin gelatin hydrolysate. J. Agric. Food Chem., 2005, 53, 581–587.
Page 14, Line 313-315
Similar results reported by Yi et al. [33] showed that soybean-derived peptides inhibited MG132-induced apoptosis of RAW264.7 cells in a dose-dependent manner by flow cytometry.
Page 26 Line 564-567
Yi, G., Li, H., Li, Y., Zhao, F., Ying, Z., Liu, M., Zhang, J., Liu, X. The protective effect of soybean protein‐derived peptides on apoptosis via the activation of PI3K‐AKT and inhibition on apoptosis pathway. Food Sci. Nutr., 2020, 8, 4591–4600.
Page 17, Line 372-374
Similar results reported by Zhao et al. [40] indicated that sea cucumber ovum peptide NDEELNK had a protective effect on mitochondrial energy metabolism disorder caused by scopolamine damage to PC12 cells.
Page 27, Line 89-592
Zhao, Y., Dong, Y., Ge, Q., Cui, P., Sun, N., Lin, S. Neuroprotective effects of NDEELNK from sea cucumber ovum against scopolamine-induced PC12 cell damage through enhancing energy metabolism and upregulation of the PKA/BDNF/NGF signaling pathway. Food Funct., 2021, 12, 7676–7687.
Page 19, Line 408-411
Similarly, Ren et al. [18] proved that the hazelnut protein-derived peptide LDAPGHR exerts anti-inflammatory effects by inhibiting LPS-induced activation of NF-κB and MAPK pathways in RAW264.7 macrophages.
Point 5: The English usage must be brushed up.
Answer: Thanks for your comment. We have ensured that the revised manuscript was rewritten a native speaker of the language to avoid grammar mistakes (revised portions are underlined in red) and provided a certificate of English editing.
Page 2, Line 26
Tricholoma matsutake peptides
Page 3, Line 50
proposed as contributing to disease progression
Page 3, Line 51-52
an imbalance in intracellular antioxidant enzymes is conducive to the accumulation of reactive oxygen species (ROS)
Page 4, Line 77
Ser-Asp-Leu-Lys-His-Phe-Pro-Phe (SDLKHFPF)
Page 6, Line 116-121
followed by pretreatment with different concentrations of peptides (50, 100, and 200 μg/mL) for 2 h and then co-incubation with LPS (1 μg/mL) for 24 h. The peptide replaced by complete medium was the blank control, while LPS was selected as a positive control. Further, 50 μL of cell media was collected and incubated with 50μL of Griess reagent I and Griess reagent II for 15 min. The OD value was read at 540 nm. The NO concentration was calculated with a standard curve from sodium nitrite.
Page 6, Line 124-127
followed by pretreatment with different concentrations of peptides (50, 100, and 200 μg/mL) for 2 h and then co-incubation with LPS (1 μg/mL) for 24 h. The peptide replaced by complete medium was the blank control, while LPS was selected as a positive control.
Page 6, Line 128-129
Enzyme Linked Immunosorbent Assay (ELISA) kits
Page 6-7, Line 133-135
followed by pretreatment with the respective peptides (200 μg/mL) for 2 h and then co-incubation with LPS (1 μg/mL) for 24 h. The peptide replaced by complete medium was the blank control, while LPS was selected as a positive control.
Page 7, Line 142-144
followed by pretreatment with the respective peptides (200 μg/mL) for 2 h and then co-incubation with LPS (1 μg/mL) for 24 h. The peptide replaced by complete medium was the blank control, while LPS was selected as a positive control.
Page 7, Line 153-155
followed by pretreatment with the respective peptides (200 μg/mL) for 2 h and then co-incubation with LPS (1 μg/mL) for 24 h. The peptide replaced by complete medium was the blank control, while LPS was selected as a positive control.
Page 8, Line 162-165
and incubated for 24 h, followed by pretreatment with the respective peptides (200 μg/mL) for 2 h and then co-incubation with LPS (1 μg/mL) for 24 h. The peptide replaced by complete medium was the blank control, while LPS was selected as a positive control.
Page 8-9, Line 176-178
followed by pretreatment with the respective peptides (200 μg/mL) for 2 h and then co-incubation with LPS (1 μg/mL) for 24 h. The peptide replaced by complete medium was the blank control, while LPS was selected as a positive control.
Page 9, Line 189-192
and incubated for 24 h, followed by pretreatment with the respective peptides (200 μg/mL) for 2 h and then co-incubation with LPS (1 μg/mL) for 24 h. The peptide replaced by complete medium was the blank control, while LPS was selected as a positive control.
Page 10, Line 208
Tricholoma matsutake-derived Peptides
Page 11, Line 233
We found that these levels were significantly higher
Page 11, Line 245
Tricholoma matsutake-derived Peptides
Page 11-12, Line 247-251
Peptides and other biological substances can alleviate the immune dysfunction caused by oxidative stress by up-regulating the intracellular antioxidant enzyme system [20]. In addition, Dia and de Mejia [21] identified a 43-amino acid peptide in soybeans and demonstrated that it could counteract chemically induced inflammation by enhancing the antioxidant defence in macrophages.
Page 12, Line 262-272
Numerous studies have shown that ROS increase dramatically following the stimulation of the cell membrane of phagocytes, which in turn triggers inflammation through a series of oxidative stress-affected signaling transduction pathways [16,22]. The critical factors that affect the physiological activity of different peptides are amino acid composition, amino acid sequence, and chemical properties such as hydrophobicity [23]. Similar results were reported by Jiang et al., [24] concluding that Leu contributes a stronger antioxidant activity than Ile does in the peptide sequence (not C-terminus or N-terminus). Mendis et al. [25] reported that the anti-oxidative potency of peptides containing Leu has been attributed to its long aliphatic side chain group that is conceivably capable of interaction with acyl chains of susceptible fatty acids.
Page 13, Line 274
Tricholoma matsutake-derived Peptides
Page 13, Line 276
Excessive accumulation of ROS impairs mitochondrial function.
Page 13, Line 290
Zhang et al. [27] reported
Page 14, Line 297
Tricholoma matsutake-derived Peptides
Page 14, Line 304-305
upon SDIKHFPF and SDLKHFPF pretreatment, respectively.
Page 14, Line 312
- matsutake peptide treatment had the opposite effect,
Page 14, Line 313-316
Similar results reported by Yi et al. [33] showed that soybean-derived peptides inhibited MG132-induced apoptosis of RAW264.7 cells in a dose-dependent manner by flow cytometry. In summary,
Page 15, Line 319
- matsutake peptide treatment had the opposite effect,
Page 16, Line 355-356
compared with the model cells.
Page 17, Line 365
which might impair the ETC
Page 17, Line 372-374
Similar results reported by Zhao et al. [40] indicated that sea cucumber ovum peptide NDEELNK had a protective effect on mitochondrial energy metabolism disorder caused by scopolamine damage to PC12 cells.
Page 17, Line 380
Effect of Tricholoma matsutake-derived Peptides
Page 18, Line 390
upregulation of IKKβ and p-IκB-α
Page 18, Line 399
significantly improved the
Page 19, Line 408-411
Similarly, Ren et al. [18] proved that the hazelnut protein-derived peptide LDAPGHR exerts anti-inflammatory effects by inhibiting LPS-induced activation of NF-κB and MAPK pathways in RAW264.7 macrophages.
Page 20, Line 435-440
In brief, we demonstrated that T. matsutake-derived peptides SDLKHFPF and SDIKHFPF attenuate the inflammatory response of LPS-induced RAW264.7 macrophages by blocking the NF-κB/COX-2 signaling pathway, exhibiting a protective effect against cell mitochondrial dysfunction. Therefore, T. matsutake peptide can be used as a potential natural food source for reducing the severity of inflammatory diseases.
Once again, thank you very much for your attention to this manuscript. We are more than willing to answer any of your questions concerning this paper and our research.

Reviewer 2 Report
In the submitted paper protective effects and molecular mechanisms of action of two T. matsutake-derived peptides, SDLKHFPF 25 and SDIKHFPF, on lipopolysaccharide (LPS)-induced mitochondrial dysfunction and inflammation in RAW264.7 macrophages are described. Research is well planned and described.
Reviewer has several suggestions for improvement:
- In 77 line hyphen (-) is missing Ser-AspLeu-Lys-His-Phe-Pro-Phe
- Figure 1 should be described in more detail (correct the legend and introduce A, B, C, D, E labels).
- Fig 6C isn't mentioned in the Results and discussion
- In the conclusion the last sentence is too general, it should be elaborated in more detail based on the presented results and derived research conclusions.
Author Response
Response to Reviewer 2 Comments
Thanks so much for your advice and guidance on our manuscript. We want to express our deep appreciation. We have revised our manuscript closely and carefully, the revised parts are listed below.
Point 1: In 77 line hyphen (-) is missing Ser-AspLeu-Lys-His-Phe-Pro-Phe
Answer: Thanks for your comment. We are so sorry for making such mistakes. We added a missing hyphen (-) in the revised manuscript.
Page4, Line 77
Ser-Asp-Leu-Lys-His-Phe-Pro-Phe (SDLKHFPF)
Point 2: Figure 1 should be described in more detail (correct the legend and introduce A, B, C, D, E labels).
Answer: Thanks for your comment. We have corrected the legend of Figure 1 in the revised manuscript.
Page 33, Line 625-627
The different capital letters represent significant difference for SDIKHFPF (P < 0.05). The different lowercase letters represent significant difference for SDLKHFPF (P < 0.05).
Point 3: Fig 6C isn't mentioned in the Results and discussion
Answer: Thanks for your comment. We are so sorry for making such mistakes. We deleted Figure 6C in the revised manuscript. In fact, Figure 6C is a graphical abstract.
Point 4: In the conclusion the last sentence is too general, it should be elaborated in more detail based on the presented results and derived research conclusions.
Answer: Thanks for your comment. We have rewritten the conclusions.
Page 20, Line 435-440
In brief, we demonstrated that T. matsutake-derived peptides SDLKHFPF and SDIKHFPF attenuate the inflammatory response of LPS-induced RAW264.7 macrophages by blocking the NF-κB/COX-2 signaling pathway, exhibiting a protective effect against cell mitochondrial dysfunction. Therefore, T. matsutake peptide can be used as a potential natural food source for reducing the severity of inflammatory diseases.
Once again, thank you very much for your attention to this manuscript. We are more than willing to answer any of your questions concerning this paper and our research.

Round 2
Reviewer 1 Report
According to the reviewer's suggestions and comments, the authors revised the manuscript. The experimental description is precisely reported to follow up the procedure.
The discussion section also improved to combine one experimental result with another one.
The reviewer pointed out the the difference of two oligopeptide sequence, namely the difference of Leu and Ile. These two amino acids are the group of BCAA (Val, Leu, Ile), it is quite reasonable to consider the big difference for the hydrophilicity, hydrophobicity and other physical properties. In the comments to the reviewer, the authors wrote the following "page 12, line 265-272 --- Leu contributes a stronger antioxidant activity than Ile does in the peptide sequence." As pointed out above, the difference of Leu and Ile is the position of the methyl group of the carbon chain. It is quite difficult to think the significant difference of the antioxidant activity present. In this meaning, it is reasonable to prepare the another BCAA (Val) for the compound.
If the authors insist on the difference of Leu and Ile, the authors (at least) show the three dimensional structure of both compounds. (molecular calculation) It does not take much time, and the authors get important information.
The reviewer also pointed out the preparation of similar oligopeptide for the comparison. It might be future study for authors.
Author Response
Response to Reviewer
Thanks so much for your advice and guidance on our manuscript. We want to express our deep appreciation. We have revised our manuscript closely and carefully, the revised parts are listed below.
Question: The reviewer pointed out the difference of two oligopeptide sequence, namely the difference of Leu and Ile. These two amino acids are the group of BCAA (Val, Leu, Ile), it is quite reasonable to consider the big difference for the hydrophilicity, hydrophobicity and other physical properties. In the comments to the reviewer, the authors wrote the following "page 12, line 265-272 --- Leu contributes a stronger antioxidant activity than Ile does in the peptide sequence." As pointed out above, the difference of Leu and Ile is the position of the methyl group of the carbon chain. It is quite difficult to think the significant difference of the antioxidant activity present. In this meaning, it is reasonable to prepare the another BCAA (Val) for the compound. If the authors insist on the difference of Leu and Ile, the authors (at least) show the three dimensional structure of both compounds. (molecular calculation) It does not take much time, and the authors get important information. The reviewer also pointed out the preparation of similar oligopeptide for the comparison. It might be future study for authors.
Answer: Thanks for your comment. We deleted the inaccurate discussion in lines 265-272 and rewritten this part in the revised manuscript to make it clearer.
Page 12, Line 262-272
The critical factors that affect the physiological activity of different peptides are amino acid composition, amino acid sequence, and chemical properties such as hydrophobicity [22]. Similar results were reported by Qian et al., [23] concluding that the antioxidative activity of a peptide depends on its molecular size and chemical properties, such as the hydrophobicity and electron transfer capacity of the amino acid residues in the sequence. Numerous studies have shown that ROS increase dramatically following the stimulation of the cell membrane of phagocytes, which in turn triggers inflammation through a series of oxidative stress-affected signaling transduction pathways [16,24]. Thus, the synthetic peptides SDIKHFPF and SDLKHFPF protect cells from highly reactive oxidation products which can cause damage to biomolecules.
Page 24-25 Line 526-536
Jiang, H. P., Tong, T. Z., Sun, J. H., Xu, Y. J., Zhao, Z. X., Liao, D. K. Purification and characterization of antioxidative peptides from round scad (Decapterus maruadsi) muscle protein hydrolysate. Food Chem., 2014, 154, 158–163.
Qian, Z. J., Jung, W. K., Byun, H. G., Kim, S. K. Protective effect of an antioxidative peptide purified from gastrointestinal digests of oyster, Crassostrea gigas against free radical induced DNA damage. Bioresource Technol., 2008, 99, 3365–3371.
Furthermore, your points and suggestions will be helpful for our future research. In this study, we preliminarily explored the protective effects and molecular mechanisms of two T. matsutake-derived peptides, SDLKHFPF and SDIKHFPF, on LPS-induced mitochondrial dysfunction and inflammation in RAW264.7 macrophages. Both peptides can effectively inhibit the activation of NF-κB/COX-2 and may confer an overall protective effect against LPS-induced cell damage. Next, we will further compare the preparation of similar oligopeptides, systematically investigate the structure-activity relationships of bioactive peptides using bioinformatics, molecular docking and Pharmcophore, and enhance their physiological activities through structural modifications.
Once again, thank you very much for your attention to this manuscript. We are more than willing to answer any of your questions concerning this paper and our research.
